# EFFICIENT MULTIVARIATE BANDIT ALGORITHM WITH PATH PLANNING

## ABSTRACT

In this paper, we solve the arms exponential exploding issues in multivariate Multi-Armed Bandit (Multivariate-MAB) problem when the arm dimension hierarchy is considered. We propose a framework called path planning (TS-PP) which utilizes decision graph/trees to model arm reward success rate with m-way dimension interaction, and adopts Thompson sampling (TS) for heuristic search of arm selection. Naturally, it is quite straightforward to combat the curse of dimensionality using a serial processes that operates sequentially by focusing on one dimension per each process. For our best acknowledge, we are the first to solve Multivariate-MAB problem using graph path planning strategy and deploying alike Monte-Carlo tree search ideas. Our proposed method utilizing tree models has advantages comparing with traditional models such as general linear regression. Simulation studies validate our claim by achieving faster convergence speed, better efficient optimal arm allocation and lower cumulative regret.

## 1 INTRODUCTION

Multi-Armed Bandit (MAB) problem, is widely studied in probability theory and reinforcement learning which dates back to clinical trial studies by Thompson Thompson (1933). Robbins Robbins (1952) formulated the setting in 1952: it includes a learner having $K$ arms (options/choices) to explore given little knowledge about the properties of each arm. At each step $t$ in $[1, T]$, the learner chooses an arm $i$ and receives a reward $X^t$ from the choice under the purpose to minimize the regret as well as maximize cumulative reward. Binomial bandit is the most common bandit formats by restricting the rewards being binary ($X^t \in \{0, 1\}$). The solution of MAB problem involves balancing between acquiring new knowledge (exploration) and utilizing existing knowledge (exploitation), to make arm selection at each round $t$ based on the state of each arm. The upper confidence bound (UCB) algorithm was demonstrated as optimal solution to manage regret bound in the order of O(log(T))Lai & Robbins (1985)Lai et al. (1987)Agrawal (1995)Auer et al. (2002). In online experiment, Thompson Sampling (TS) algorithm attracts a lot of attention due to its simplicity at implementation and resistance in batch updating. TS algorithm for binomial bandit could achieve optimal regret bound as well Kaufmann et al. (2012).

Many modern online applications (e.g. UI Layout) have configuration involving multivariate dimensions to be optimized, such as font size, background color, title text, module location, item image etc., each dimension contains multiple options Hill et al. (2017) Nair et al. (2018). In this paper, we call it Multivariate-MAB problem. The exploring space faces exponentially exploding number of possible configurations as dimensions are added into the decision making. TS algorithm is reported to convergences slowly to the optimal solution Hill et al. (2017) when dealing with Multivariate-MAB problem. To speed up convergence, one common enhanced TS solution is to model the expected reward as general linear model (TS-GLM) Chu et al. (2011)Bubeck et al. (2012)Scott (2010) by probit/logit link function with $m$-way dimension interaction features. TS-GLM gives up the ability to fit certain complex interactions, in exchange for focusing on lower-dimensional parameter space and achieves better solution. However, updating derived posterior sampling algorithm in TS-GLM demands imputing the multivariate coefficients and creates computation burden at each iteration Scott (2010)Scott (2015)Hill et al. (2017). To release such burden, Daniel et. al. Hill et al. (2017) proposed Hill-climbing multivariate optimization Casella & Berger (2002) for TS-GLM, and recognized it obtained faster convergence speed with polynomial exhaustive parameter space.

Different from TS-GLM, our proposal framework called "Path Planning" (TS-PP) is quite straight-forward to combat the curse of dimensionality by a serial processes that operates sequentially and focuses on one dimension at each component process. Further more, it treats arm reward naturally with m-way dimension interaction by m-dimensional joint distribution. Our novelty includes: **(a)** modeling arm selection procedure under tree structure. **(b)** efficient arm candidates search strategies under decision graph/trees. **(c)** remarkable convergence performance improvement by straightforward but effective arm space pruning. **(d)** concise and fast reward function posterior sampling under beta-binomial even with $m$-way dimension interaction consideration. Compare to TS-GLM, TS-PP avoids deriving complex and slow posterior sampling in GLM, while still effectively leveraging the $m$-way dimension interactions and achieving even better performance by reducing arm space with efficient search strategies.

This paper is organized as follows: We first introduce the problem setting and notation; then we explain our approach in details, and further discuss the differences among several variations; we also examine the algorithm performance in simulated study and concludes at the end.

## 2 MULTIVARIATE-MAB PROBLEM FORMULATION

We start with the formulation of contextual multivariate MAB: the sequential selection decision of layout (e.g. web page) $A$, which contains a template with $D$ dimensions and each dimension contains $N$ options, under context $C$ (e.g. user preference) for the purpose of minimizing the expected cumulative regret $(R)$.

For each selected layout $A$, a reward $X$ shall be received from the environment. Here only binary reward $(X \in \{0, 1\})$ is discussed, but our approach can be extended to categorical/numeric rewards as well. In the layout $A$ template, there exists $N_i$ alternative options for dimension $i$ and $f_i$ is denoted as the selected option. We further assume $N_i = N$ in our following description for simplicity purpose. The chosen layout $A$ can be denoted as $[f_1, \ldots, f_D] \in \{1, \ldots, N\}^D$, and it utilizes the notation $A[i]$ referring $f_i$. Context $C$ includes extra environment information that may impact layout's expected reward (e.g. device type, user segments, etc.).

At each step $t = 1, \ldots, T$, the bandit algorithm selects arm $A^t$ from $N^D$ search space with the consideration of the revealed context $C^t$ in order to minimize the cumulative regret over $T$ rounds:

$$R_T = \sum_{t=1}^{T} (E(X|A^{\star t}, C^t) - E(X|A^t, C^t))$$

where $A^{\star t}$ stands for the best possible arm at step $t$. Generally, $R_T$ is on the order $O(\sqrt{T})$ under linear payoff settings Dani et al. (2008)Chu et al. (2011)Agrawal & Goyal (2013). Although the optimal regret of non-contextual Multivariate MAB is on the order $O(logT)$ Lai & Robbins (1985). In this paper, we focus on categorical-contextual multivariate MAB where $C$ are purely categorical variables. By solving multivariate MAB independently for each combination of $C$ (assuming not too many), it is trivial to show that the optimal regret bound of $R_T$ is still $O(log(T))$. Without loss of generalization, we set context feature as constant and ignore $C$ in the following discussion.

## 3 RELATED WORK

### 3.1 PROBABILISTIC MODEL FOR MULTIVARIATE-MAB

To model the multivariate bandit reward of layout $A^t$, we denote the features combining $A$ and interactions within $A$ (possibly non-linearly) as $B_A \in R^M$ with length $M$. The $B_A$ could involve only upto $m$-way dimension interactions of $A$ $(O(N^m))$ instead of capturing all $(O(N^D))$ possible interactions. The linear model with pairwise interactions is as follows:

$$B_A^\top \mu = \mu^0 + \sum_{i=1}^{D} \mu_i^1(A) + \sum_{i=1}^{D} \sum_{j=i+1}^{D} \mu_{i,j}^2(A) \tag{1}$$

where $\mu$ are fixed but unknown parameter coefficients. In above function, it contains common bias term $\mu^0$, weights for each dimension of layout $\mu_i^1(A)$ and 2-way dimension interactions of layout $\mu_{i,j}^2(A)$. The sub-indexes $i$ and $j$ are referring dimension $i$ and $j$ correspondingly.

Under the GLM setting, $g(P(X = 1|B_A)) = g(p) = B_A^\top \mu$, where $p$ is the success rate of reward $X$ and $g$ is the link function that can either be the inverse of normal CDF $\Phi^{-1}(p)$ as a probit model or the $ln(\frac{p}{1-p})$ as a logit model. For given $B_A$, the likelihood of reward $X$ would be $(\Phi(B_A^\top \mu))^X (1 - \Phi(B_A^\top \mu))^{(1-X)}$ or $(\exp(-B_A^\top \mu) + 1)^{-X} (\exp(B_A^\top \mu) + 1)^{(-1+X)}$ for probit or logit model respectively. The posterior sampling distribution of reward is its likelihood integrates with some fixed prior distribution of weights $\mu$. Updating the posterior, at step $t$, requires solving GLM for $\mu$ from cumulative historical rewards $H^{t-1} = [(X^1, A^1), \ldots, (X^{t-1}, A^{t-1})]$, which is disturbing and creates computation burden with time.

Daniel et. al. Hill et al. (2017) proposed **MVT2** by assuming probit model with interactions between dimensions (Equation 1) and employing Hill-climbing multivariate optimization to achieve faster convergence speed.

## 3.2 THOMPSON SAMPLING

Thompson sampling (TS) Russo et al. (2018) is widely adapted in solving bandit and reinforcement learning problems to balance between exploitation and exploration. It utilizes common Bayesian techniques to form posterior distribution of rewards, hence allocates traffic to each arm proportional to probability of being best arm under posterior distribution.

Normally we handle binary response as binomial distribution with Beta prior $\texttt{Beta}(\alpha_0, \beta_0)$ to form posterior distribution $\texttt{Beta}(\alpha_k + \alpha_0, \beta_k + \beta_0)$, where $\alpha_k$ and $\beta_k$ are the number of successes and failures it has been encountered so far at arm $k$, as well as $\alpha_0$ and $\beta_0$ are prior parameters and would been set as 1 for uniform prior. At selection stage in round $t$, it implicit allocates traffic as follows: simulates a single draw of $\theta_k$ from posterior ($\texttt{Beta}(\alpha_k + \alpha_0, \beta_k + \beta_0)$) for each arm $k$ and the arm $k^* = \arg\max_k(\theta_k)$ out of all arms will be selected. At update stage, it collects reward $X^t \in \{0, 1\}$ and the reward is used to update hidden state $(\alpha_{k^*}, \beta_{k^*})$ of selected arm.

Practically, to solve Multivariate-MAB problem, algorithm $\mathbf{N^D - MAB}$ directly adopts TS in selection out of $N^D$ arms, while algorithm **D-MABs** decomposes Multivariate-MAB into $D$ (dimensions) sub-MABs and implements TS in selection out of $N$ arms for each MAB (dimension) independently. We would discuss more details of the two algorithms in following sections.

## 3.3 MONTE-CARLO TREE SEARCH

Monte-Carlo Tree Search (MCTS)Browne et al. (2012)Chaslot et al. (2008a) is a best-first heuristic search algorithm to quickly locate the best leaf node in a tree structure. In game tree problem, it achieves great success especially when the number of leafs is large. Generally, each round of MCTS consists of four steps Chaslot et al. (2008b): selection, expansion, simulation and back-propagation. Essentially, a simplified case only need selection and back-propagation steps. At selection step of MCTS, it starts from root of the tree and adopts a selection policy to select successive child nodes until a leaf node $L$ is reached. The back-propagation step of MCTS involves using the reward to update information (like hidden states) in the nodes on the path from selected leaf to root. In artificial intelligence literature, the most successful (MCTS algorithm) UCT utilizes UCB Kocsis & Szepesvári (2006) as its node selection policy. Applying TS as node selection policy in MCTS (TS-MCTS) is not well investigated in literature Imagawa & Kaneko (2015) from our best knowledge.

By introducing the hierarchical dimensional structure of bandit arms, we can build tree structure over bandit arms and deploy MCTS with TS techniques for arm selection. We prefer TS node selection policy due to its resistance in performance for batch update situation. Inspired by this idea, we establish TS Path Planning algorithm to solve Multivariate-MAB problem.

## 4 APPROACH

In this paper, we propose TS path planning algorithm (TS-PP) for Multivariate-MAB problem to overcome the exponential explosion in arm searching space of $\mathbf{N^D - MAB}$ algorithm. Stimulated by MCTS idea, we utilize similar heuristic search strategy to locate best arm under a tree structure. We call such tree structure as "decision tree" which is constructed purely by $D$ dimensions. Notably there are $D!$ decision trees constructed in different sequential dimension order over same leaf nodes

and they assemble a "decision graph". Under a decision tree/graph, the arm selection procedure is decomposed into a serial processes of decision making that operates sequentially and focuses on value selection within one dimension per each process. At each sequential decision process, we would apply TS as successive child nodes (dimension value) selection policy. The sequential order of dimensions ("decision path") is determined by the path planning strategy.

Figure 1 shows an example of decision graph, decision tree as well as decision path. Without loss of generality, we assume that dimensions are tagged in order of $[d_1 : d_D]$, which is an arbitrary order of $[1, \ldots, D]$. Decision tree in Figure 1b compactly represents the joint probabilities of all arms $[f_{d_1:d_D}]$ (leaf nodes) and internal nodes. Here we borrow notations in Kochenderfer (2015). $[d_1 : d_D]$ and $[f_{d_1:d_D}]$ are compact ways to write $[d_1, \ldots, d_D]$ and $[f_{d_1}, \ldots, f_{d_D}]$ respectively. The sub-index $d_i$ of $f_{d_i}$ refers that this dimension value $f_\star$ comes from dimension $d_i$. The structure of decision tree consists of nodes and directed edges. Each level of the decision tree represents a dimension $d_i$ and each node at that level corresponds to a value $f_{d_i}$ for dimension $d_i$. Directed edges connect parent node to child node where the arrow represents conditional (jointly) relationship. Associated with each node $f_{d_i}$ is a jointly probability $P(f_{d_i}, \text{prePath}(f_{d_i}))$, where $prePath(f_{d_i})$ represents all predecessor nodes of $f_{d_i}$ in red arrows (Figure 1b). We denote $prePath(f_{d_i})$ as $f_{d_1:d_{i-1}}$ in our example, and $path(f_{d_i})$ as $f_{d_1:d_i}$. Based on the chain rule, the likelihood of arm $[f_{d_1:d_D}]$ is $P(f_{d_1:d_D}) = \prod_{i=1}^{D} P(f_{d_i}|f_{d_1:d_{i-1}}) \propto P(f_{d_i}|f_{d_1:d_{i-1}}) \propto P(f_{d_1:d_i})$. In practice, $P(f_{d_1:d_i})$ could be represented by hidden states $(\alpha, \beta)$ from Beta distribution (binary rewards) for node $f_{d_i}$ (given its prePath($f_{d_i}$)) and node's states could be updated in back-propagation stage (like in MCTS). As in TS, the chance of arm $f_{d_1:d_D}$ being the best arm depends on $P(f_{d_1:d_D})$, hence it is also partially related with $P(f_{d_1:d_i})$. Instead of sampling directly from posterior distributions of $N^D$ arms, sampling from distribution associated with each node $f_{d_i}$ could also provide guidance on value selection for dimension $d_i$. Figure 1a utilizes decision graph to compactly represent $D!$ decision trees. Once a decision path (red arrow in Figure 1a) is determined, decision graph is degenerated to decision tree for detailed view. With such abstraction, we further extended the naive MCTS idea with several other path planning strategies.

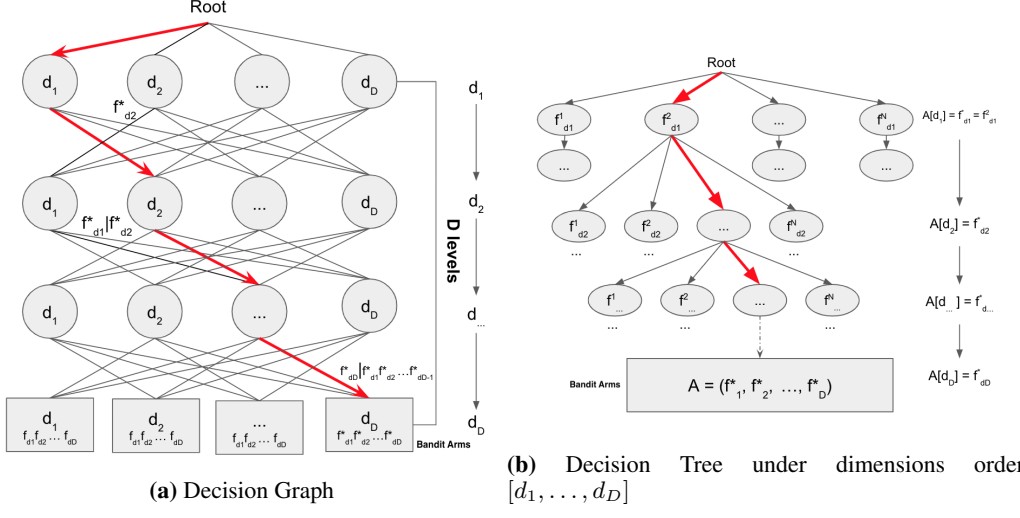

**(a)** Decision Graph

**(b)** Decision Tree under dimensions order $[d_1, \ldots, d_D]$

**Figure 1:** Path Planning Overview

## 4.1 TS-PP Template

Algorithm 1 provides TS-PP template for better understanding of our proposal over big picture. The proposed path planning algorithms utilize different path planning strategies to obtain candidate arm, by navigating the path from one node to next node originating from start to destination of the decision graph and applying TS within selected node (dimension) to pick the best value per each dimension, condition on fixing selected multivariate options in predecessor nodes unchanged. This conditional posterior sampling distribution as mentioned previously would rely on the hidden states of dimension values within current node given predecessor nodes dimensional value choices.

---

**Algorithm 1** TS-PP Template

---

1: **Input** D, N, $S \in R^1$, $\alpha_0 = 1$, $\beta_0 = 1$
2: **for** step $t = 1, 2, \ldots$ **do**
3:     **for** search $c = 1 \to S$ **do**                        ▷ Candidates Construction Stage
4:         $A^c \leftarrow$ Path Planning Procedure
5:         Sample $\theta^c \leftarrow$ **Sample**$(A^c)$
6:     Select Arm $A^t \leftarrow \arg\max_{A^c}(\theta^c)$
7:     Update History Rewards $\mathbf{H}^t \leftarrow \mathbf{H}^{t-1} \cup (X^t, A^t)$
8: **function** SAMPLE(Path=$[f_{d_1:d_L}]$ )
9:     Get $\alpha, \beta$ from **Node**$(f_{d_L}|f_{d_1:d_{L-1}})$
10:     Sample $\theta \sim$ **Beta**$(\alpha + \alpha_0, \beta + \beta_0)$
11:     **Return** $\theta$

---

We understand that searching candidates in this way might be stacked in sub-optimal arms. To address this issue, we intentionally repeat our candidate searching for $(S)$ times and re-apply TS tricks among these $S$ candidates for final arm selection. Once the arm is chosen at step $t$, we would back-propagate the collected reward $X^t$ to update the hidden states of nodes (**Node**$(f_{d_L}|f_{d_1:d_{L-1}})$) within all the possible paths from selected leaf to root in decision trees. Here the notation **Node**$(f_{d_L}|f_{d_1:d_{L-1}})$ is used to load the node $f_{d_L}$'s (with prePath $f_{d_1:d_{L-1}}$) hidden states in to memory, which corresponds the joint density $P(f_{d_1:d_L})$. It worth to note that any relative order of $[f_{d_1:d_L}]$ represents the same joint distribution (with same hidden states). In practice, **Node**$(f_{d_L}|f_{d_1:d_{L-1}})$ requires $O(1)$ computation complexity. But it could also be implemented in $O(T)$ computation complexity for lazy back-propagation with cache memory saving.

## 4.2   PATH PLANNING PROCEDURE

We propose four path planning procedures for candidates searching: Full Path Finding (**FPF**), Partial Path Finding (**PPF**), Destination Shift (**DS**) and Boosted Destination Shift (**Boosted-DS**). To construct arm-candidate under decision graph (figure 1a), **FPF** starts from the root and sequentially optimizes dimensions one-by-one in a completely random order, which utilizes the depth-first search (DFS) strategy. With sticking with top-down flavor while extending the **D-MABs**, **PPF** utilizes the breath-first search (BFS) strategy with $m-$dimensional joint distribution independence (explained later) for all $m$-sub dimensions out of $D$. Finally, inspired by hill-climbing Casella & Berger (2002) Hill et al. (2017), which start from a random initial arm (bottom node in decision graph) and optimize value for one dimension with all other dimension value fixed, such bottom-up flavor **DS** and the advanced version **Boosted-DS** would be discussed. The following explains four methods in details.

**Full Path Finding FPF** is the direct application of MCTS and describes DFS algorithm in graph search. Starting from top, **FPF** randomly picks a permutation of $D$ dimensions denoting as $[d_{1:D}]$ with equal chance to construct a decision tree, and recursively applies TS policy from nodes on the path from root to leaf in that decision tree. It follows dimensions order to sequentially optimize value $f_{d_i}^\star$ for each dimension $d_i$. The upper index $\star$ refers the optimized value for target dimension $d_i$. Since we repeat FPF $S$ times, each iteration $c$ picks different decision tree (permutation of $D$ dimensions) and construct one candidate $A^c$. The computational complexity for full path finding is $O(SND)$ with $S$ times searching and space complexity is $O(N^D)$. For lazy back-propagation implementation, the computation and space complexity could be improved to $O(SNDT)$ and $O(SDT)$ separately.

**Partial Path Finding** In contrast, **PPF** describes a kind of BFS algorithm. A $m^{\text{th}}$-partial path finding (**PPF**$m$) recursively applies TS policy from nodes on pre-path $[d_{1:(m-1)}]$ up to level $m-1$ in decision graph, then it simultaneously visits the remaining $(D - m + 1)$ dimensions (un-visited nodes) in parallel at level $m$ and apply TS policy correspondingly. Specifically, the **D-MABs** method is equivalent to **PPF1**, which adopts the dimension independent assumption. The Pseudo code in Algorithm 2 between line 5 and 10 illustrates a **PPF2** algorithm, which assumes pairwise joint distribution independence. Mathematically, variables $A$ and $B$ are conditionally independent give $C$ ($A \perp B|C$) if and only if $P(A, B|C) = P(A|C)P(B|C)$, and would call joint distribution of $(A, C)$ and $(B, C)$ are independent. So pairwise dimensional joint distribution independence means

---

**Algorithm 2** Path Planning Procedures

---

1: **procedure** FULL PATH FINDING
2:     **for** index $i = 1 \rightarrow D$ in (random order) $[d_{1:D}]$ **do**
3:         $f^*_{d_i} \leftarrow \textbf{TS}(\text{tgtDim}=d_i, \text{prePath}=[f^*_{d_1:d_{i-1}}])$ and $A^c[d_i] = f^*_{d_i}$
4:     Constructed Candidate $A^c$
5: **procedure** PARTIAL PATH FINDING
6:     Random Pick Dimension $d_i \in [d_{1:D}]$
7:     $f^*_{d_i} \leftarrow \textbf{TS}(\text{tgtDim}= d_i, \text{prePath}= \emptyset)$ and $A^c[d_i] = f^*_{d_i}$
8:     **for** index $(j = 1 \rightarrow D)$ and $(j \neq i)$ **do**
9:         $f^*_{d_j} \leftarrow \textbf{TS}(\text{tgtDim}=d_j, \text{prePath}=[f^*_{d_j}])$ and $A^c[d_j] = f^*_{d_j}$
10:    Constructed Candidate $A^c$
11: **procedure** DESTINATION SHIFT
12:    Initial $A^c$ = random layout $A$
13:    **for** $k = 1 \rightarrow K$ **do**
14:       Random Pick Dimension $d_i \in [d_{1:D}]$
15:       $f^*_{d_i} \leftarrow \textbf{TS}(\text{tgtDim}=d_i, \text{prePath}=A^c[-d_i])$ and $A^c[d_i] = f^*_{d_i}$
16:    Constructed Candidate $A^c$
17: **procedure** BOOSTED DESTINATION SHIFT
18:    Initial $A^c$ = random layout $A$
19:    **for** $k = 1 \rightarrow K$ **do**
20:       Random Pick Dimension $d_i \in [d_{1:D}]$
21:       $f^*_{d_i} \leftarrow \textbf{bstTS}(\text{tgtDim}=d_i, \text{prePath}=A^c[-d_i])$ and $A^c[d_i] = f^*_{d_i}$
22:    Constructed Candidate $A^c$
23: **function** TS(tgtDim = $d_L$, prePath=$[f_{d_1:d_{L-1}}]$ )
24:    **for** $f = 1 \rightarrow N$ **do**
25:       $\theta^f \leftarrow \textbf{Sample}(\text{Path}=[f_{d_1:d_{L-1}}, f_{(d_L)}])$
26:    **Return** $f^*_{d_L} \leftarrow \arg\max_f(\theta^f)$
27: **function** BSTTS(tgtDim = $d_L$, prePath=$[f_{d_1:d_{L-1}}]$ )
28:    **for** $f = 1 \rightarrow N$ **do**
29:       $\mu^1_{d_L} \leftarrow \textbf{Sample}(\text{Path}=[f_{d_L}])$
30:       **for** index $i = 1 \rightarrow L - 1$ **do**
31:          $\mu^2_{d_i,d_L} \leftarrow \textbf{Sample}(\text{Path}= [f_{d_i}, f_{d_L}])$
32:       $\theta^f \leftarrow \mu^1_{d_L} + \sum_{i=1}^{L-1} \mu^2_{d_i,d_L}$
33:    **Return** $f^*_{d_L} \leftarrow \arg\max_f(\theta^f)$

---

that $d_{(1:D)_{-i}} \perp |d_i$ for $\forall i \in (1 : D)$, where $d_{(1:D)_{-i}} \perp$ stands for all dimensions are independent except for dimension $d_i$. Intuitively, **PPF2** assumes pairwise interactions between dimensions, as it draws samples from pairwise dimensional joint distribution. Generally **PPF**$m$ maps up to $(m)$-way interactions in regression model. The optimal computational complexity of **PPF2** is $O(SND)$ with $S$ times searching and space complexity is $O(ND^2)$, if we only load all hidden states from top 2 levels of decision graph into memory.

**Destination Shift DS** randomly picks an initial arm (bottom node in decision graph) and performs Hill-climbing method cycling through all dimensions for $K$ rounds. At each round $k$, we randomly choose an dimension $d_i$ to optimize and return the best dimension value $f^*_{d_i}$ based on posterior sampling distribution condition on the rest of the dimension values prefixed ($A^c[-d_i]$). We then use $f^*_{d_i}$ to generate $A^c_{(k)}$ from $A^c_{(k-1)}$ by $A^c_{(k-1)}[d_i] = f^*_{d_i}$. The computational complexity is $O(SNK)$ and space complexity is $O(N^D)$.

**Boosted Destination Shift Boosted-DS** utilizes **bstTS** instead of **TS** function for value optimization on each target dimension node $d_i$. It extends our previous intuition that sampling from $m$-dimensional joint distribution is 1-to-1 mapping to m-way interaction weights in regression model. Pseudo code in Algorithm 2 between line 17 and 22 describes 2nd-Boosted-DS (**Boosted-DS2**) sampling strategy which follows equation 1 with pairwise $(2 - way)$ interaction assumption. Instead of single draw from arm $P(A^c[-d_i], f_{d_i})$, at round $k$ with target dimension $d_i$, it sums samples drawing from 1-way joint density $(P(f_i))$ and all pairwise $(2 - way)$ joint distributions with $f_{d_i}$ $(P(f_{d_i}, f_{d.})$ for all sub-index $\cdot \in [1 : D]$ and $\cdot \neq d_i)$. Generally, $m^{\text{th}}$-Boosted-DS (**Boosted-DSm**) would take the sums of drawing samples upto $m-$dimensional joint distributions. The computa-

tional complexity of **Boosted-DS2** is $O(SKND)$ and space complexity is $O(ND^2)$ if we store all needed hidden states instead.

In summary, **FFP** utilizes hidden states from same decision tree at each iteration; **PFP** and **Boosted-DS** only utilizes hidden states on top levels of decision graph; **DS** utilizes hidden states on leaf nodes. **DS** and **Boosted-DS** randomly pick an layout $A$ to start and keep improving itself dimension by dimension till converge, while **FFP** and **PFP** do not randomly guess other dimension values. All four algorithm approximate the process of finding best bandit arm by pruning decision search trees and greedy optimization of sequential process through all dimensions. As the greedy approach significantly reduce search space, hence the converge performances are expected to beat the traditional Thompson sampling method $\mathbf{N^D - MAB}$.

## 5 EMPIRICAL VALIDATION AND ANALYSIS

We illustrate the performance of our algorithm (**FPF**, **PPF**, **DS** and **Boosted-DS**) on simulated data set, comparing with **MVT**Hill et al. (2017), $\mathbf{N^D - MAB}$Hill et al. (2017) and **D-MABs**Hill et al. (2017) base models mentioned before. Specially, we evaluate (a) the average cumulative regret, (b) the convergence speed and (c) the efficiency of best optimal arm selection among these models under same simulation environment settings. To access fairly appreciative analysis, the mechanism and parameters for generating the simulation data set are completely at random. We would also replicate all algorithms multiple (H) times and take the average to eliminate evaluation bias due to TS probabilistic randomness. Furthermore, we extensively exam the cumulative regret performance of proposed algorithms by varying (1) the relative strength of interaction between dimensions and (2) complexity in arm space (altering $N$ and $D$) to gain comprehensive understanding of our model.

### 5.1 SIMULATION SETTINGS

Simulated reward data is generated in Bernoulli simulator with success rate being linear with $m$-way dimension interactions:

$$\Phi(\theta) = \frac{1}{\beta}[\mu^0 + \alpha_1 \sum_{i=1}^{D} \mu_i^1(A) + \cdots + \alpha_m \sum_{d_1=1}^{D} \cdots \sum_{d_m=d_{m-1}+1}^{D} \mu_{d_1,\ldots,d_m}^m(A)] \qquad (2)$$

where $\beta$ is scaling variable and $\alpha_1$, ..., $\alpha_m$ are control parameters. It is trivial to set $\mu^0 = 0$. We intentionally generate weights $\mu$ independently from $N(0,1)$, and set $\beta = m$ and $\alpha_m = \frac{m!}{D(D-1)\ldots(D-m+1)}$ to control the overall signal to noise ratio as well as related strength among $m$-way interactions.

In this paper, we set $m = 2$ (pairwise dimension interaction), $D = 3$ and $N = 10$ in above simulator settings, which yields 1000 possible layouts. To observe the convergence of each model and eliminate the randomness, our simulation is generated with $T = 100,000$ time steps and $H = 100$ replications. On each simulation replica and at time step $t$, layout $A^t$ is chosen by each algorithm, and a binary reward $X^t$ is sampled from Bernoulli simulator with success rate $\theta$, which is coming from Equation 2 with pre-generated randomly weights $\mu$. We choose $S = 45$ and $K = 10$ as the same Hill climbing model parameter settings to compare between **FPF**, **PPF2**, **DS**, **Boosted-DS2** and **MVT2** methods.

### 5.2 NUMERICAL RESULTS

Figure 2 shows the histograms of arm exploration and selection activities for 7 algorithm as well as distribution of success rate for arms in our simulator. The horizontal axis is the success rate of selected arm, while the vertical axis is the probability density in histogram. The success rate density of Bernoulli simulators is symmetrically distributed, which coincides with our simulation setting. However, the severity of right skewness reveals the efficiency and momentum algorithms recognizing bad performance arms and quickly adjusting search space to best possible arms. Although $\mathbf{N^D}$-**MAB** is theoretically guaranteed to archive optimal performance in the long run, the histogram graph might empirically explains why **MVT2**, **FPF**, **PPF2** and **Boosted-DS2** outperform $\mathbf{N^D}$-**MAB** in many ways. It is worth to mention that the search behavior (performance) of **DS** is similar to $\mathbf{N^D}$-**MAB**, but **DS** consists simpler computational complexity ($O(SNK)$). This concludes us that **DS**

strategy itself starting path planning from bottom has limited improvement on arm heuristic search than $\mathbf{N^D}$-**MAB**. The underline reason could be that only small fraction of arms is explored at early stage and little information is known for each arm, starting from top strategy can resemble dimensional analogues and learn arm's reward distribution from other arms with similar characteristics. In turns, it helps to rapidly shift to better performed arms. The proposed **Boosted-DS2** overcomes **DS**'s issue by using TS samples from top levels. The heavily right skewness in **Boosted-DS2** histogram confirms our suspects.

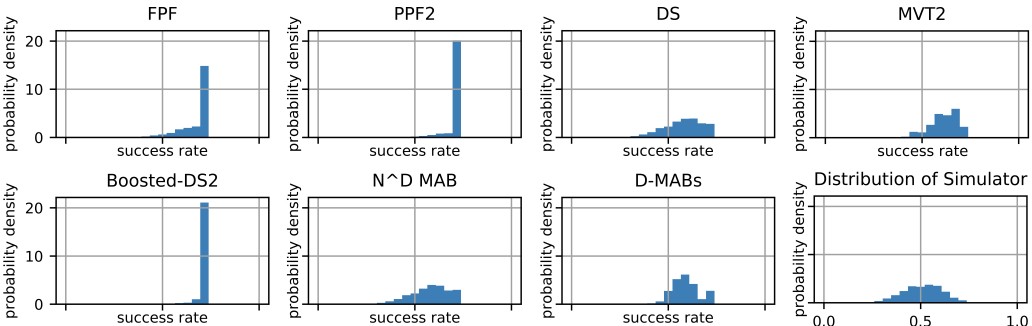

**Figure 2:** Histogram of expected reward for historical arm search.

| Algorithm | FPF | PPF2 | DS | Boosted-DS2 | MVT2 |
|---|---|---|---|---|---|
| Iteration Speed (it/s) | 7.04 | 22.39 | 2.01 | 1.38 | 0.25 |

**Table 1:** Algorithm Speed Comparison

To recognize the effectiveness of optimal selection, we leverage the average regret, convergence rate and best arm rate. We define convergence rate as proportion of trials with the most selected layout over a moving window with batch size $(t_1 - t_0 =)$ 1000. We further specify best arm rate as the proportion of trials with the best possible layout in one batch.

$$\textbf{Average Regret} = \sum_{h=1}^{H} \frac{1}{H} \sum_{t=1}^{T} \frac{E(X_{A^\star}) - X^t}{T}, \quad \textbf{Convergence Rate}[t_0, t_1] = \sum_{h=1}^{H} \frac{1}{H} \sum_{t=t_0}^{t_1} \frac{\mathbf{1}_{A^\circ}(A^t)}{t_1 - t_0},$$

$$\textbf{Best Arm Rate}[t_0, t_1] = \sum_{h=1}^{H} \frac{1}{H} \sum_{t=t_0}^{t_1} \frac{\mathbf{1}_{A^\star}(A^t)}{t_1 - t_0},$$

where $A^\circ$ and $A^\star$ stand for most often selected layout and best possible layout respectively within a batch. Ideally, we prefer convergence rate and best arm rate both approaching to 1, which means the algorithm converges selection to single arm (convergence) as well as best arm. In practice, a fully converged batch trials almost surely select the same layout (sub-optimal) but not necessarily the global optimal layout.

Simulated performance results are displayed in Figure 3 where x-axis is the time steps. Path planning algorithms demonstrate advantages over base models, especially for **FPF**, **PPF2** and **Boosted-DS2**. We see that **PPF2** and **Boosted-DS2** quickly jump to low regret (and high reward) within 5000 steps, followed by **FPF** and **MVT2** around 10000 steps. Although **Boosted-DS2** and **MVT2** share fastest convergence speed followed by **PPF2** then **FPF**, but **FPF** holds the highest best arm rate. **FPF** performance in cumulative regret (and reward) catches up for longer iterations as well. The intuition behind these is that **FPF** includes the most complex model space with considering full dimension interactions, in which it not only look from top level of decision graph to quickly eliminate bad performed dimension values but also drill down to leaf arms to correct negligence from higher levels. The exponential space complexity or computational complexity proportional to $T$ is our concern of **FPF** compared with **PPF2** and **Boosted-DS2**.

In our experiment, **PPF2**, **Boosted-DS2** and **MVT2** assume models with pairwise interactions in one way or the other, and it happens to be our simulator setting. In practice extra effort is needed for correct modeling the reward function, which is out of this paper's scope. **PPF2** and **Boosted-DS2** both efficiently achieve lower regret comparing with **MVT2**. However, **PPF2** carries better best arm rate than **Boosted-DS2**. Our take-away from this is that Hill-climbing strategy contains two drawbacks. First, it is equivalent to a bottom-up path planning strategy in our framework, which

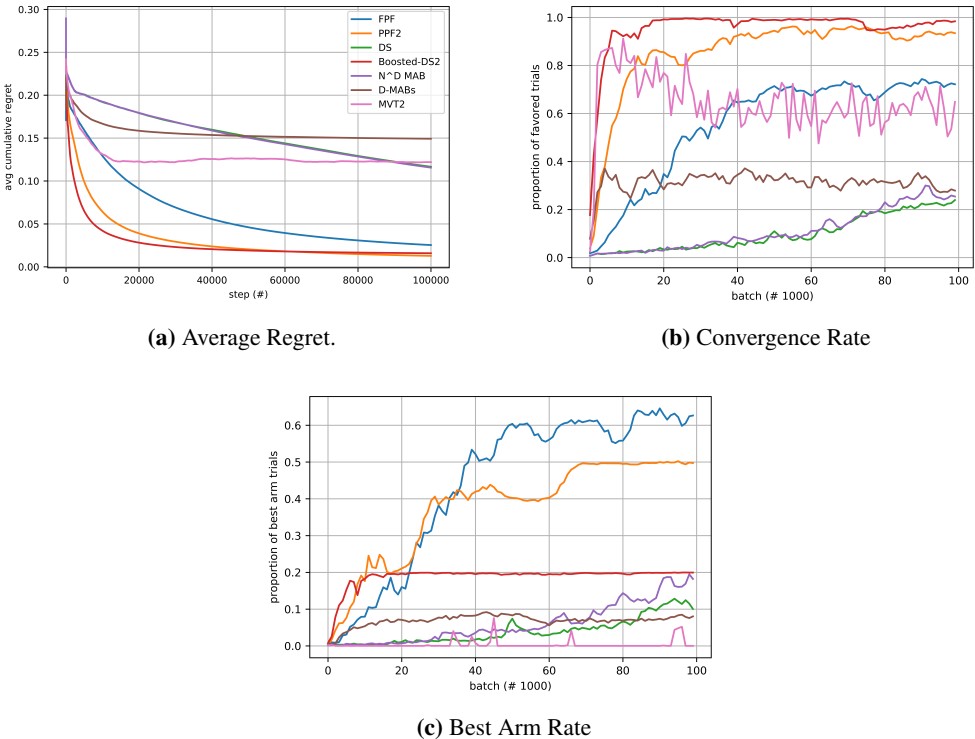

**(a)** Average Regret.

**(b)** Convergence Rate

**(c)** Best Arm Rate

**Figure 3:** Performance on simulated data with $D = 3, N = 10, \beta = 3, m = 2, \alpha_1 = \frac{1}{3}$ and $\alpha_2 = \frac{1}{3}$.

is not as efficient as top-down strategy as discussed before. **Boosted-DS2** combats such weakness using TS samples on top levels to mimic sample draw from lower level. Second, Hill-climbing starts with randomly guess other dimension values which easily ends up with good enough arm selection (low regret and high convergence) but not always the best (low best arm rate). In the meanwhile, **D-MABs** struggles in performance as its assumption of independence between dimensions doesn't match with our simulator.

Although **PPF2**, **Boosted-DS2** and **MVT2** all share simplified model complexity (both in computation and parameter space), however **MVT2** takes longer time period per iteration compared with the other two. Table 1 shows the iteration speed of these algorithms in our implementation. In fact, **MVT2** is the slowest algorithm as the heavy computation burden when updating regression coefficients in posterior sampling distribution.

We further extend our simulation results of average cumulative regret by varying $\alpha_2$ to change the strength of interactions as well as varying $N$ and $D$ to change space complexity in Fig 4. We skip **MVT2** due to the time limitation (**MVT2** takes 5 days per experiment). As $\alpha_2$ varies from $\frac{1}{6}$ to 1 with step size $\frac{1}{6}$, we see the pattern consists with prior result at Fig 4a. The only exception is **D-MABs** gets dominant regret when interaction strength is weak ($\alpha_2 = \frac{1}{6}$), as **D-MABs**'s no interaction assumption close to the truth at that time. **D-MABs** is equivalent with **PPF1**. So **D-MABs** should perform similarly with **PPF2** when interaction strength is weak. Next we analyze the impact on performance with model complexity. We systematically swap $N$ in $(2, 4, 6, 8, 10, 12)$ and $D$ in $(2, 3, 4, 5)$ at Fig 4b and 4c respectively. We observe that the relative performance still holds: **FPF** $\simeq$ **PPF2** > **Boosted DS** > **DS** $\simeq$ **N$^D$-MAB** > **D-MABs**. Based on these extensive experiments, we assert that our proposed method is superior consistently.

In summary, our simulated results suggest that TS-PP has good performance overall for multivariate bandit problem with large search space when dimension hierarchy structure exists. **FPF** accomplishes the best performance, however **PPF2** attracts implementation attention due to its computation efficiency with comparable performance.

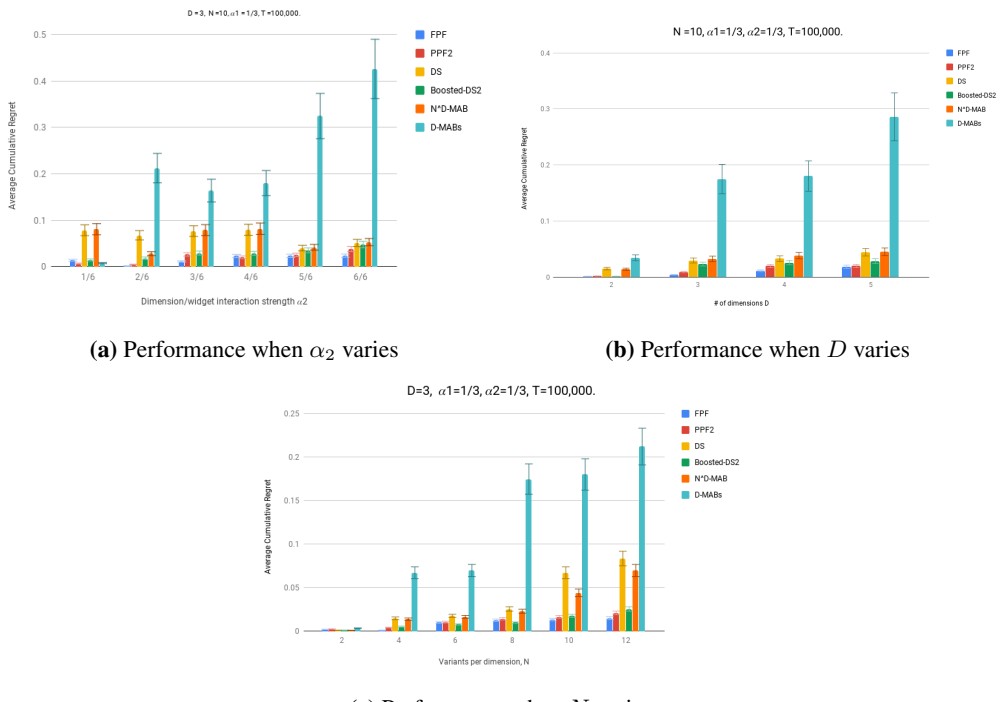

**(a)** Performance when $\alpha_2$ varies

**(b)** Performance when $D$ varies

**(c)** Performance when $N$ varies

**Figure 4:** Cumulative Averaged Regret over number of iterations when D, N and $\alpha_2$ varies.

## 6 CONCLUSIONS

In this paper, we presented TS-PP algorithms taking advantage of the hierarchy dimension structure of bandit arms to quickly find the best arm. It utilizes decision graph/trees to model arm reward success rate with m-way dimension interaction, and adopts TS with MCTS for heuristic search of arm selection. Naturally, it is quite straightforward to combat the curse of dimensionality using a serial processes that operates sequentially by focusing on one dimension per each process. Based on our simulation results, it achieves superior results in terms of cumulative regret and converge speed comparing with **MVT**, **N$^D$-MAB** and **D-MABs** on large decision space. We listed 4 variations of our algorithm, and concluded that **FPF** and **PPF** conduct the best performance. We highlight **PPF** due to its implementation simplicity but high efficiency in performance.

It is trivial to extend our algorithm to contextual bandit problem with finite categorical context features. But how to extend our algorithm from discrete to continuous contextual variables worth us further exploration. We notice some related work of TS-MCTS Bai et al. (2014) dealing with continuous reward in this area. Finally, fully understanding the mechanism of using heuristic greedy approach (in our method) to approximate TS from $N^D$ arms is still under investigation.

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
