# OpenReview forum: "Efficient Multivariate Bandit Algorithm with Path Planning"
_ICLR.cc/2020/Conference — Reject_

### Official Review · AnonReviewer1 · 2019-10-24
**Official Blind Review #1**

**Rating:** 1

**Review:**

This paper proposes to use path planning on the tree structure for multivariate MAB problem. The idea is to model the hierarchical structure in the decision space, thus tree search idea can be applied to improve the efficiency of learning.

In general the trend of the paper is easy to follow. However, the proposed method is a bit ad-hoc and I don’t think the paper provides enough justification for it.
1. FPF is not really practical in that it needs to main D! decision trees, each one with N^D leaves. On the other hand, other approximation methods like PPF, relies on some independence assumption. However, this paper provides not analysis about such independence even in simulations. For example, when these independence assumptions are strongly violated, what the performances would be.
2. There is no experiments on real data. The simulation in the this paper is very specific and I am not sure how representative it could be. The simulation results don’t provide a clear message too.

Other comment
1. On page 4, why would P(f_{d1:d_D}) be propotional to P(f_{d1:d_i})?


=========================
After Rebuttal
I don't think the rebuttal addresses my concern: the proposed method is not well justified and its performance is not really analyzed nor with convincing evidence.

**Experience Assessment:**

I have published one or two papers in this area.

**Review Assessment: Checking Correctness Of Derivations And Theory:**

I assessed the sensibility of the derivations and theory.

**Review Assessment: Checking Correctness Of Experiments:**

I did not assess the experiments.

**Review Assessment: Thoroughness In Paper Reading:**

I read the paper at least twice and used my best judgement in assessing the paper.

---

> ### Author Response · Authors · 2019-11-15
> **about your main questions**
>
> Thank you for your comments.
>
> 1. We understand that FPF needs to maintain D! trees. But comparing with maintaining N^D arms, it is not that bad. We further provide other approximation methods like PPF by assuming up to m-way interactions, which is normal in our practice of fitting regression models.
>
> 2. Our simulation covariates are randomly generated and we also control its relavent weights by changing alpha and beta.

---

### Official Review · AnonReviewer3 · 2019-10-24
**Official Blind Review #3**

**Rating:** 6

**Review:**

This paper approaches the problem of exploding arms in multivariate multi-armed bandits. To solve this problem, the authors suggest an approach that uses Thompson sampling for arm selection and decision trees and graphs (inspired from Monte-Carlo tree search) to model reward success rates. They propose to versions of this path planning procedure and 2 version inspired from Hill-climbing methods. They validate their claims on a simulation showing better performance and faster convergence, and providing an extensive analysis of the results.

The idea seems novel. The paper is well structured, but the writing can be improved, and some parts are hard to read and follow (See minor comments below).

Here are few questions:
- The authors claim that the approach can be extended to categorical/numeric rewards. Can they give more details on how?
- The experiments are done only with D=3 and N=10. How easy would it be to scale to higher dimensions?
- In the curves of Figure 3, N^D-MAB and DS seem not converged yet contrarily to the other methods. Have the authors tried to let them run for longer to see to which values they converge?

Minor comments (non-exhaustive examples):
- Punctuation issues:
    *Multi-Armed Bandit (MAB) problem, is widely ...
    * Generally, RT is on the order O( T) under linear payoff settings Dani et al. (2008)Chu et al. (2011)Agrawal & Goyal (2013). Although the optimal regret of non-contextual Multivariate MAB is on the order O(logT ) ...

- Imprecise statements:  ... for each combination of C (assuming not too many), ...

- Sentences to rewrite:
     * The posterior sampling distribution of reward is its likelihood integrates with some fixed prior distribution of weights μ.
     * ..., and would call joint distribution of (A, C) and (B, C) are independent.
     * ..., which means the algorithm converges selection to single arm (convergence) as well as best arm.

- Typos: Jointly distribution/relationship -> joint ,It worth to note -> it is worth to note, extensively exam -> extensively examine, a serial processes -> process, would been -> be ...


**Experience Assessment:**

I do not know much about this area.

**Review Assessment: Checking Correctness Of Derivations And Theory:**

I did not assess the derivations or theory.

**Review Assessment: Checking Correctness Of Experiments:**

I assessed the sensibility of the experiments.

**Review Assessment: Thoroughness In Paper Reading:**

I read the paper at least twice and used my best judgement in assessing the paper.

---

### Official Review · AnonReviewer2 · 2019-10-31
**Official Blind Review #2**

**Rating:** 3

**Review:**

In this paper, the authors address the curse of dimensionality in Multivariate Multi-Armed Bandits by framing the problem as a sequential planning problem.
Their contributions are the following:
1. They propose to solve the Multivariate MAB sequentially across each dimension, which I think is a reasonable and not very common approach in the literature compared to e.g. structured bandits, even though the authors themselves call this approach "natural" and "quite straightforward".
2. They propose a framework based on a Monte-Carlo Tree Search procedure to solve this sequential decision problem.
3. They introduce several approximations and heuristics to avoid traversing the entire tree and trade-off performance for computational complexity, which they evaluate empirically.

The obtained results seem to support their original claim: the tree-based approaches (e.g. FPF) perform better than the considered baselines. However, this claim is at the same time contradicted by the fact that a depth-one planning procedure with heuristic leaf evaluation (PPF2) reaches similar performances, which indicates that the underlying problem can be solved greedily and does not involve long-term (m-way) interations and limits the interest of the main idea in this paper. This is probably due to the design of experiments which only model pairwise interactions, but then this questions the relevance of the proposed empirical evaluation with respect to the claim.
Generally, though several good ideas were presented, I felt that they were not properly motivated and that the paper generally lacks rigor and clarity. The many variants and modeling decisions proposed seem quite arbitrary, and in the end fail to provide any valuable insight.

Several aspects of the proposed approach and algorithms were not clear or justified. I will list some of them, grouped by the corresponding contribution:

1. Multivariate MAB framed as a planning problem.
No intuition is provided regarding why it is expected to perform any better than classical N^D-MAB. In this absence of assumption regarding additional structure to the problem considered (such as, say, the independence assumption in D-MABS), is there any reason to think so? (for instance, all the N^D arms are present at the leaves of the tree) The authors only mention in their conclusion that they leverage a "hierarchy dimension structure", but this claimed structure was is never defined nor formalized in the paper. I can only guess that such a structure would involve a particular ordering of dimensions such that shallow ones have more influence on the payoff than the deeper ones? But then, it would be unknown in practice, and the choice of the particular ordering used in TS-PP is expected to have a crucial influence on its performance. Yet, the sensitivity of their algorithms to the ordering is never discussed by the authors -they choose it arbitrarily-  nor evaluated in their experiments.

2. The proposed TS-PP framework
- The authors propose using MCTS with TS instead of UCB as a sampling rule, which they claim is quite novel: "Applying TS as node selection policy is not well investigated from our best knowledge". However, the authors do not mention important reference (Bai et al, 2014) until the very end of the paper in their conclusion: "We notice some related work dealing with continuous rewards in this area". Yet, Bai et al. also consider Bernoulli rewards in their experiments and use conjugate Beta priors, exactly like the authors of the present work do. Moreover, the choice of TS rather than UCB is not properly motivated, and the UCT algorithm is not considered as a baseline in the experiments. Hence, there is no basis to assess whether TS is particularly relevant or not for this problem.
- They also propose to repeat S times the candidate searching procedure in Algorithm 1 in order not to be "[stuck] in sub-optimal arms". This property is normally ensured by optimistic sampling rules, it is not clear why it is required here. More importantly, the sensitivity of the Algorithm 1 to S is not evaluated.
- I did not fully understand the Candidates Construction Stage in Algorithm 1. For instance, in the case of FPF, for each search c=1..S (l3) there is a call (l4) to Full Path Finding() to generate a candidate A^c. From what I understand, a random ordering is chosen (Algorithm 2, l2), and a single rollout is performed (l2, the dimensions 1->D are iterated from root to leaf) to return a candidate. When we move to the next search c -> c+1, since a new ordering d_1:D is chosen then the previous tree storing the models for p(fd_i | fd_1...fd_i-1) is not relevant anymore and must be discarded. Does that mean that the priors at the nodes are recomputed for each search/ordering/tree given the entire history, instead of being maintained and updated? The authors do state that "FFP utilizes hidden states from the same decision tree", but I do not see how this is feasible when the dimensions ordering changes at each iteration.
- This confusion is related to the fact that the update of the Beta posterior does not appear anywhere in Algorithm 1 and 2. This is unfortunate, especially considering that contrary to TS in a normal MAB setting, the observations for an internal node are not i.i.d. since they depend on the underlying planning procedure, which induces a distributional drift as more sample are collected and probably make the derived posterior invalid. Has this issue been considered by the authors?

3. The variants of FPF
The idea behind Partial-PF-m is common in the tree-search literature: to stop at early depths and use a heuristic to estimate the value at the leaf. Here, the suggestion in the particular context of Multivariate-MAB to use TS with an independence assumption (D-MABs) is interesting and sound. However, It requires the knowledge the degree m of interactions, whereas the idea of MCTS is precisely to avoid using such heuristics for leaf evaluation by replacing them without random rollouts. Other methods such as DS start at a leaf and move locally which rather resembles Hill-climbing more closely than actual Tree-Search, and I do not really see how they fit in the TS-PP framework?

Finally, I found the manuscript difficult to read due to many language mistakes and typos.

Minor comments:
- The citation style is incorrect.
- What is the point of introducing contextual MABs in the problem formulation when the context is not going to play any part whatsoever in the presented work? This only brings unnecessary complexity and impedes clarity.
- In the experiments, H=100 replications are performed, which is appreciated, but unfortunately the corresponding confidence intervals or standard deviations are not shown, only the mean performance.
- I found the section 3.1 unclear. Is B_A^T \mu a dot product? Where do the coefficients of B_A or its length M appear in (1)? How is M chosen, how is B_A obtained?
- Figure 1.a is also unclear, the graph seems to be used to define permutations/orderings of dimensions 1..D. But all paths are not valid (eg. d1 -> d2 -> d1), ad random orderings are used in Algorithm 2, so where is this graph used exactly?


=============
After Rebuttal:
On one hand, some of the points mentioned are now clearer (e.g. 1 & 2.2, 2.3).
On the other hand, most of my concerns remain valid, and some were not addressed.
Typically, the dependency to an unknown (and undefined...) optimal ordering and the underlying optimization procedure (random search) are not properly discussed, whereas they seem central to justify the tree-based approach.
Other remaining concerns are more generally the lack of justification and clarity of the proposed framework and implemented variants, and the inadequacy of the experiments with respect to the claim (using m=2 basically means only the first depth of the tree is useful). Therefore, I will maintain my initial score.

**Experience Assessment:**

I have published one or two papers in this area.

**Review Assessment: Checking Correctness Of Derivations And Theory:**

N/A

**Review Assessment: Checking Correctness Of Experiments:**

I assessed the sensibility of the experiments.

**Review Assessment: Thoroughness In Paper Reading:**

I read the paper at least twice and used my best judgement in assessing the paper.

---

> ### Author Response · Authors · 2019-11-15
> **about your main questions**
>
> Thank you for your comments.
>
> 1, about your first concern that sensitive to the ordering. In our algorithm, we do not just rely on one particular ordering of dimensions, instead, we randomly pick multiple different dimensions orders from all the permutation of the dimensions orders. We repeated S times to go through different dimension ordering to overcome the sensitivity of just using one dimensions order.
>
> 2.1, Regarding Bai etal 2014: In Bai’s paper, it utilize MAB algorithm UCT to solve Markov Decision Process. However as we said in the beginning of our paper, we are interested in solving MAB problem inspired by Monte Carlo Tree Search. especially, in online experiment, Thompson Sampling (TS) algorithm attracts a lot of attention due to its simplicity at implementation and resistance in batch updating.
>
> 2.2, about the repeat S times the path planning procedure in Algorithm 1, this part is very necessary. after each run of path planning procedure, we only find one sub-optimal arm based on the heuristic dimension ordering. we will have S different candidates arms and then select the best one within the S arms to roll out using Thompson sampling.
>
> 2.3, about the question on the "node" priors updating. Here the notation Node(fdL|fd1:dL−1) is used to load the node fdL’s (with prePath fd1:dL−1) hidden states in to memory, which corresponds the joint density P(fd1:dL). It worth to note that any relative order of [fd1:dL] represents the same joint distribution (with same hidden states).  So, in practice, we do not need to recompute them every time but just maintain all the joint density P(fd1:dL). And then, the Node(fdL|fd1:dL−1) will only requires O(1) computation complexity without the need of re-computation.
>
> 2.4, we use back propagation to update Beta posterior,  which is trivial for our setting. The posterior of an internal node does not only depend on itself but also the previous planning procedure, that’s why we use Node(f_dL | f_d1 : f_d(L-1)) to difference them.
>
> 3, for the variants of path planning methods, actually we considered different assumptions, the degree m of interactions is important knowledge/assumptions when using BOOSTED DESTINATION SHIFT method, the independence assumption is for D-MAB, and the conditional independence assumption is for PPF. for the DS, it comes from hill climbing idea, however, we can also treat it as bottom-up tree-search.

---

### Decision · Program_Chairs · 2019-12-19

**Decision:**

Reject

**Comment:**

This paper tackles the multivariate bandit problem (akin to a factorial experiment) where the player faces a sequence of decisions (that can be viewed as a tree) before obtaining a reward. The authors introduce a framework combining Thompson Sampling with path planning in trees/graphs. More specifically, they consider four path planning strategies, leading to four approaches. The resulting approaches are empirically evaluated on synthetic settings.

Unfortunately, the proposed approaches lack theoretical justification and the current experiments are not strong enough to support the claims made in the paper. Given that most of reviewer's concerns remained valid after rebuttal, I recommend to reject this paper.